# Can the home experience in luxury hotels promote pro-environmental behavior among guests?

**Meixin Liu[1], Xingxing Peng[2]***

1 School of Management, Guangdong Polytechnic Normal University, Guangzhou, China, 2 College of Tourism, Hunan Normal University, Changsha, China

* peng_xingxing2018@163.com

## Abstract

While the home is an important place for individuals to act pro-environmentally, researchers have rarely explored the pro-environmental behavior of hotel customers in terms of their home away from home experiences during their travels. This study uses a combination of qualitative (interviews) and quantitative (questionnaires) methods to explore customer experiences of home spaces in the hotel context and the relationship between people's experience in hotels and their pro-environmental behavior. The study shows that (1) customers' experience of home spaces in hotels occurs through three dimensions: the function of home, the emotion of home, and the imagination of home. (2) Both the function of home and the emotion of home exert a significantly positive impact on hotel customers pro-environmental behavior. (3) The imagination of home exerts a significant positive effect on pro-environmental behavior both inside and outside of the hotel. (4) The pro-environmental behavior of customers in their own homes has a positive moderating effect on the relationship between the home experience and pro-environmental behavior in the hotel context. By combining the concepts of home spaces and pro-environmental behavior, this study, on the one hand, bridges the research gap between place experience and pro-environmental behavior in the hotel context; on the other hand, the study transcends the limitations engendered by studying pro-environmental behavior in the hotel and home space from a binary perspective.

## 1 Introduction

Environmental crises such as frequent climate extremes and resource depletion have become more severe, and hotels have significantly contributed to climate warming [1]; therefore, sustainable development has become an important goal in hotel management. With the rapid growth of the modern hospitality industry, the consumption of products and services by customers has increased significantly [2]. Moreover, the luxury hospitality industry has been continuously committed to improving customers' experiences, possibly leading to the further draining of resources. Although the hotel industry has strived to implement energy-efficient

**Data Availability Statement:** All relevant data are within the paper and its Supporting Information files.

**Funding:** Financial support recipient: Peng Xingxing Source of funding: The present study was

funded by National Natural Science Foundation of China (Grant No. 42101232); Natural Science Foundation of Guangdong Province (Grant No. 2022A1515010740).

**Competing interests:** Declaration of competing interest: The authors report no conflicts of interest.

and emission-reducing solutions, it has not been sufficient [3]; for hotels to truly transition into green hotels, these efforts need to be combined with pro-environmental behavior on the part of customers.

In a hedonistic context, the high-quality experience of a luxury hotel can present a great challenge for the demonstration of pro-environmental behavior by customers. luxury hotels are the benchmark of the accommodation industry, and they have the highest level of facilities and service management. In the tourism context, luxury hotels are a "home away from home" for customers, and they also strive to create a home form and home atmosphere [4], e.g., high-end furniture and appliances, warm and meticulous care, etc., s to provide customers with an ideal home experience. However, the high-quality experience provided by luxury hotels is likely to lead to the excessive consumption of resources by customers and thus cause more serious environmental problems. It is worth noting that for customers, on the one hand, a luxury hotel serves as another home during their journey, and home is the most important to practice pro-environmental behaviors [5]. On the other hand, it has been shown that customers are willing to engage in environmentally friendly consumption and use their own initiative to minimize damage to the environment [6, 7]. Therefore, the question becomes whether customers act pro-environmentally, as they do at home, when hotels provide them with a home experience. This question is central to this study.

The relationship between tourists' experiences and their pro-environmental behavior has been repeatedly demonstrated [8–10], but few studies have focused on hotel contexts. First, the tourism experience is a combination of feelings and reactions that arise from the interaction between the tourist and the external environment during the tourism process; this interaction includes perceptions, emotions and meaningful experiences [11], specifically entertainment, education, escape and aesthetics [10]. This is also true in the hotel context, but the specific contents of tourists' experiences differ there [12]. Therefore, the pro-environmental behavior of hotel customers cannot be explored through the study of tourists' experiences. Second, of the existing relevant studies, few studies have examined the relationship between people's experiences in hotels and their pro-environmental behaviors, and those that do have generally explored customers' pro-environmental behaviors in relation to a single experience of the hotel space [2], however, customers' experiences in hotels are multidimensional. Therefore, a more thorough understanding of hotel customers' pro-environmental behaviors requires a focus on the hotel context and a multidimensional perspective on people's experience in hotels. Furthermore, a hotel essentially represents a temporary home during a journey, and it principally provides an experience of home [13], which encompasses multiple dimensions, such as the perception of physical spaces, emotional interactions, and the generation of meaning [14, 15]. It is therefore logical to consider the home experience in the hotel context as a specific touristic experience. According to the above, the objectives of this study are as follows: first, the study investigates the different dimensions of the customer's experience of home in hotels and develops a scale to measure this experience; second, the study investigates the effect of these dimensions on the pro-environmental behavior of customers; finally, semistructured interviews are conducted to further substantiate the relationship between customers' experience of home in hotels and their pro-environmental behaviors to further explore the mechanisms involved in that relationship, thus ensuring the credibility and explanatory power of the findings.

Although spatial experience plays an important role in promoting pro-environmental behaviors, the extent of its influence varies across individuals, and therefore, the threshold for the home experience to influence pro-environmental behaviors in hotel customers need to be further identified. According to existing research, pro-environmental behaviors in the home context interact with social identity, place attachment, cognitive dissonance [16] and

habituation [17] and exert a significant positive effect on pro-environmental behaviors in tourism contexts [18]. As the experience of home in a hotel context also contains the elements of cognition, identity and emotion, the pro-environmental behaviors engendered at home may likewise be a moderating variable between the experience of home and pro-environmental behavior. Thus, the third objective of this study is to examine the moderating effect of pro-environmental behaviors that occur in the home on the relationship between people's experience of home and pro-environmental behavior in the hotel context.

In summary, this study aims to explore the mechanisms by which people act pro-environmentally based on their home experience in the hotel context. Theoretically, this study aims to first construct a conceptual framework and measurement scale to characterize the home experience in a hotel context and then expand the existing research on spatial experience and pro-environmental behavior in the hotel context from a multidimensional spatial perspective, and finally to compensate for the shortcomings of exploring the pro-environmental behavior of home and that in hotels from a binary perspective. In practical terms, the findings of this study have implications for hotel managers in terms of how to effectively enhance customers' pro-environmental behaviors.

## 2 Literature review and research hypothesis

### 2.1 The experience of home in hotels

Home is a space that is both concrete and abstract, full of meaning, emotions, experiences and social relations, and it is central to human life [19]. On the one hand, the house is the basis for the construction of a home, providing a place where the practices of everyday life are enacted and a shelter from the elements and helping people to escape the dangers and disturbances of life that arise in the outside world [20]. On the other hand, the home is not limited to the house but is a 'sociospatial system' [21] that combines a physical space (the house) and social relations (the family); moreover, the home is a metaphorical space where power relations as informed by identity, class and gender, as well as interactions between people and places at different scales, play out [22]. Home is therefore a multidimensional concept, constituting not only a physical and architectural space but also a place of emotional attachment and a symbolic space [19, 23].

In the tourism context, the home experience has become a key means of attracting and retaining customers in the hospitality industry, and can be divided into three main dimensions: the function of home, the emotion of home and the imagination of home [24]. First, regarding the function of home, Young talked about home as the space people occupy, i.e., a space that is an extension of the human body and where people exhibit basic survival activities. If an individual has no place to live and perform basic life activities, then this individual is left without security or opportunity to practice basic activities [25]. Thus, the function of home spaces is to meet people's basic needs, such as eating, sleeping, storing goods and entertaining; it constitutes a place where individuals carry out family activities [26], a key place for individuals to express their identities [27], and a place of origin and return [28]. Second, regarding the emotion of home, home is an intimate space that revolves around social relations; common emotional practices include daily communication, collective celebration of festivals, and gatherings [29], resulting in emotional belonging and security, based on factors such as warmth, pleasure, solidarity, and memory. In the face of increased modernity and mobility, the function of home has gradually weakened, but the emotion of home has not faded; in fact, it has become more important [30]. Third, regarding the imagination of home, the context of postmodernity has revealed negative values about people being programmatic, conservative and meaningless; thus, the home has ceased to be a positive place where people aspire to be [31].

People try to find, create and reconstruct home in various ways; one important way lies in the act of travel. During a journey, travelers' imagination of an ideal home is fulfilled by the combination of the function of home, the emotional basis of home, a beautiful natural environment, a rich cultural atmosphere [32] and people's nostalgia for their homeland [33]. Based on this combination, this study examines how people experience hotels as 'homes' based on the function of home, the emotion of home and the imagination of home.

## 2.2 The home experience in hotels and pro-environmental behaviors

Although pro-environmental behaviors have not been universally defined and have often been used along with terms such as eco-friendly behaviors, environmentally friendly behaviors, green behaviors and environmentally responsible behaviors (ERBs) [34], in essence, pro-environmental behaviors refers to acts that aim at "consciously reducing the negative impact of one's behavior on the natural world or demonstrating environmentally beneficial behavior" [35]. In the tourism and hospitality industry, pro-environmental behaviors often include water conservation, towel reuse, energy saving, eco-product purchases, local product use, reuse of plastic bottles or bags, and reduction in food waste at tourist venues or locations [36].

According to Ballantyne et al. [37], tourism experiences can change individuals' pro-environmental behaviors through sensory impressions, emotional touch and reflective responses. For example, on the one hand, the interactions between people and places in tourism contexts provide travelers with a variety of emotional experiences such as care, recreation and escapism, which play an important role in enhancing pro-environmental behaviors [15]; moreover, aesthetic experiences in touristic destinations are also associated with emotional experiences [38]. On the other hand, tourism destinations provide opportunities for conservation and environmental education, helping customers to acquire new environmental knowledge and skills [39, 40] and promoting behavioral changes toward pro-environmental behaviors [37, 39]. When placed in a hotel context, customers interact with hotel spaces, gaining cognitive and emotional experiences that ultimately lead them to have a home experience, which in turn may change their pro-environmental behaviors. Based on this finding, this study elaborates on the hypothetical relationship between hotel-based home experiences and the pro-environmental behaviors of individuals on the following three levels: the function of home, the emotion of home and the imagination of home.

People's level of satisfaction with how much hotels make them feel at home is derived not only from the material products and cultural landscapes there that are similar to those they would find in the intimacy of their own homes but also from the identity that is expressed within these products and landscapes. On the one hand, hotels (especially luxury hotels) are committed to creating the perfect home experience for their customers in the context of a modern society and culture, based on ideals about what a dream house should be, interior designs found in magazines and ideal accommodations. This commitment satisfies people's basic needs for accommodations and food as well as their need for aesthetic experiences during their travels, prompting them to develop a relationship of functional dependence with the hotel, which in turn may lead to a higher level of environmental protection and pro-environmental behavior [41]. On the other hand, when hotel staff engage in green practices, including when they provide green products and services, they enrich their customers' environmental knowledge and educate them about the environment, which is an important part of the process of building a sense of home that not only satisfies customers' environmentalist identity but also helps to enhance their environmental awareness; thus, these efforts on the part of hotel staff motivate customers to act pro-environmentally [2].

People's emotional experience of home in a hotel is always associated with positive words such as "warmth." "pleasure" and "privacy," and this positive emotional connection is strongly connected to motivation and behaviors [9]. Specifically, when individuals feel more emotionally attached to a place, they are more willing to protect the environment and more inclined to proactively implement environmentally responsible behaviors; in turn, these behaviors directly contribute to the ecological sustainability of places and spaces. In addition, when people feel at home in hotels, they also feel empathy toward nature; this feeling often arises because hotels create green spaces for people to experience or integrate the local natural environment to create pleasant and relaxing home spaces for customers. These spaces constitute prerequisites for customers to feel empathy toward nature, which is an emotional experience that humans feel and share with nature; moreover, the more empathy people feel toward nature, the more they act in pro-environmental ways [42].

In the hotel context, customers can develop an imaginary representation of home that is based on functional satisfaction and emotional experiences but also includes experiential engagement and escapism. Both experiential engagement and escapism reflect the extent to which customers are fully immersed in the hotel spaces [43]; on the one hand, these spaces reinforce people's functional and emotional experience of home and, on the other hand, these spaces may lead people to reflect and devote their attention to specific activities that improve their environmental knowledge and behaviors [10]. These behaviors, in turn, motivate them to exhibit more pro-environmental behaviors [39].

Based on the above arguments, the following hypotheses are proposed.

H1: In the hotel context, people's satisfaction with the function of home contributes to their pro-environmental behaviors.

H2: In the hotel context, the emotion of home promotes pro-environmental behaviors.

H3: In the hotel context, the imagination of home contributes to customers' pro-environmental behaviors.

## 2.3 Moderating effects of pro-environmental behaviors at home

In the hotel context, although the experience of home may enhance customers' pro-environmental behaviors, pro-environmental behaviors vary across different customers in everyday situations, which may lead to heterogeneous behavioral responses among different individuals with the same level of experience. To explore the boundary conditions under which the experience of home influences whether customers engage in pro-environmental behaviors, the role of intimate feelings of home in pro-environmental behaviors is explored. Pro-environmental behaviors at home refer to environmentally friendly behaviors in which individuals engage at home, for example through acting in resource-efficient ways (e.g., buying energy-efficient appliances), engaging in green consumption (e.g., buying organic food), recycling renewable resources and encouraging others to adopt pro-environmental behaviors [44]. It has been shown that the spillover effects of pro-environmental behaviors in different contexts are correlated, with positive and negative effects [45]. Researchers have shown that pro-environmental behaviors that occur in the privacy of people's homes have a significantly positive effect on pro-environmental behaviors in tourism contexts [18]. The positive contextual spillover effects are often driven by social identity, place attachment and cognitive dissonance [16].

In this study, on the one hand, customers' pro-environmental behaviors occurring in their homes have been shown to promote their pro-environmental behaviors in hotels. According to social identity and cognitive dissonance theory, when individuals act in more pro-environmental ways, they come to identify as 'environmentalists' across different contexts; otherwise, they feel psychological discomfort [46, 47]. In other words, individuals who engage in pro-

environmental behaviors in a private home space believe in the effectiveness of such pro-environmental behaviors in protecting the natural environment [48], which enhances their identification with the environment, and when individuals enter a homelike situation in a hotel context, they practice pro-environmental behaviors more actively in order to maintain consistency in environmental identification [49]. In addition, MacInnes et al. [17] found that habits are an important driver of pro-environmental behavior in tourism contexts; for example, customer awareness of and concern for environmental degradation in their private home establishes good habits that are automatically expressed in tourism contexts, even while viewing the hotel as a private home, such as avoiding disposable products, saving water and energy, reusing towels, recycling, etc. [50]. On the other hand, pro-environmental behaviors conducted in private homes positively enhance the relationship between the home experience in hotels and pro-environmental behavior in hotels. When customers have a higher home experience in a hotel, it indicates that the amenities and usage habits of the hotel are similar to those of their private home; in other words, the more similar that the contexts are, more positive a positive spillover effect that the pro-environmental behavior of the private home will exert on the hotel space [51]. Furthermore, customers who act more pro-environmentally at home usually identify as environmentalists and have developed good environmental habits [18]. Even when customers do not feel at home in a hotel, their environmentalist predisposition and good environmental habits can lead them to act pro-environmentally across contexts. The converse is also true. Based on the above discussion, the following hypothesis is proposed.

H4: Customers' pro-environmental behaviors at home exert a positive moderating effect on the relationship between their home experience in a hotel and the likelihood they will act pro-environmentally within that hotel.

Based on the above research hypothesis, this study uses the theoretical framework of "experience-behavior" to clarify the influence mechanism of the home experience in a hotel (home function, home emotion and home imagination) and hotel customers' pro-environmental behavior and to explore the moderating effect of pro-environmental behavior in a private home on the relationship between hotel home experience and customers' pro-environmental behavior.

The proposed framework is depicted in Fig 1.

## 3 Research methodology

In this study, mixed methods were used, integrating qualitative and quantitative data to gain a deeper understanding of the research questions [52]. On the one hand, because the current understanding of the home experience in hotels is still at a relatively general stage, the lack of identification of the specific connotations and dimensions of the home experience makes it difficult to apply quantitative methods. On the other hand, most of the existing data on pro-environmental behavior are derived from questionnaires, but the quality of these data is often questioned and thus other types of data are needed for verification. Based on this, this study adopts a combination of qualitative and quantitative research methods. Specifically, researchers first collected and explored qualitative data through text analysis and focus groups to develop a new psychometric instrument or a new variable. In the second phase, researchers collected and tested quantitative data based on the newly developed instrument, while supplementing the qualitative interview data to further validate and interpret the quantitative analysis results.

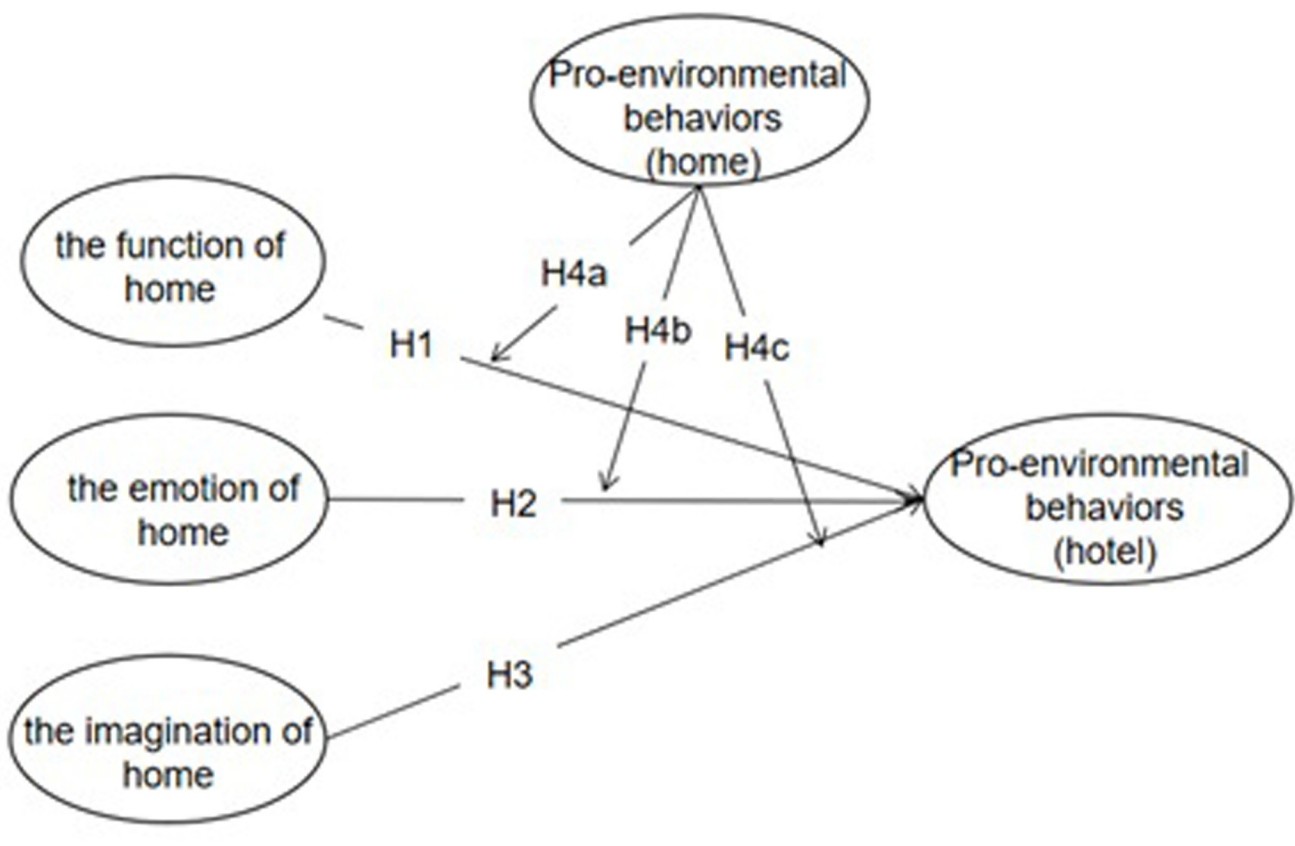

**Fig 1. Research model.**

### 3.1 Qualitative research: Exploring the different scales at which people experience hotels as homes

As no scale has been developed to measure customers' experience of home in hotels, this study used focus group discussions as the main research method to determine the scale and the content of the items in two steps. First, a focus group was organized to identify the factors that customers felt were critical for them to feel at home in a hotel. The focus group participants included six frequent guests of luxury hotels (three male and three female, with a cumulative total of 20 or more hotel days per year), three professors in tourism management-related disciplines (two in hotel management and one in tourism management), and two high-star hotel managers (with 8 or more years of hotel experience); these participants were selected because (1) they had an extensive experience with the construction of home spaces in hotels, and (2) compared to other hotels, the home experience is richer and more profound in luxury hotels. Due to the specific nature of the epidemic, the focus group discussion took the form of an online meeting where participants freely shared their opinions and feelings about the home experience in hotels. Seven factors were identified, including (1) hardware facilities in hotels (guest room amenities, food and beverage services, entertainment facilities, and meeting facilities); (2) surrounding facilities (public transportation, neighborhood restaurants and leisure and entertainment venues); (3) daily management (customer management and space management); (4) service provision (regular services and personalized services); (5) experiential activities (traditional and local cultural activities and leisure and entertainment activities); (6) a

green atmosphere (a green environment, green objects, and green spaces); and (7) a cultural atmosphere (the hotel's architectural design, brand culture, and local traditional culture). The literature has also highlighted the importance of the above factors to people's experience of hotel spaces as home spaces [13, 53–55].

Second, a categorization approach was used to further abstract the factors that were identified as influencing people's home experience in hotels. First, to avoid interferences from the outside, the categorization process was carried out by the participants individually [56]; second, after the categorization had been completed, the categories created by the different participants were compared, and any disagreements that arose were resolved through discussion. Through this process, participants agreed on a total of three dimensions. The first is the function of home, which includes (1) the amenities in the hotel, (2) the surrounding facilities, and (3) daily management. The second dimension pertains to the emotion of home, which includes (4) services and (5) experiential activities. The third dimension pertains to the imagination of home, and this dimension includes (6) a green atmosphere and (7) a cultural atmosphere.

Finally, a way to measure the home experience in hotels was developed based not only on the focus group discussions but also on the existing research pertaining to hotel service management and the home experience in hotels [13, 57]. Specifically, six questions were selected to test the functional, emotional and imaginary dimensions of home.

## 3.2 Quantitative research phase: Data collection

In this study, a questionnaire was used to collect data from the respondents. Based on findings from previous qualitative studies, a three-dimensional scale of people's home experience in hotels was summarized, with each of the three dimensions eliciting six different questions. In this study, the measurement of pro-environmental behaviors is based on the scale that Miao and Wei [58] specifically developed to compare pro-environmental behaviors in hotels and at home, although some modifications were made to apply that scale to the Chinese context. All items were measured using a seven-point Likert scale, ranging from 1 (strongly disagree) to 7 (strongly agree). In addition, the questionnaire was first pretested by academics and hospitality industry experts, revised and adjusted based on relevant feedback, and further reviewed and refined by two experts.

Four representatives of luxury hotels in China were selected as case studies for this research; they were categorized as business, conference and resort hotels. The study took place from December 2021 to January 2022 and from July 2022 to August 2022. Methods used included on-site questionnaires, and five experienced and well-trained postgraduate students were selected. The research participants were mainly selected from customers who had relaxed in the public space or checked out of the hotel because they had already experienced feeling at home in the hotel and may have demonstrated pro-environmental behaviors. In addition, during the survey, an attempt was made to reduce possible biases through process control to avoid distortion of the questionnaire data since on-site questionnaires can bring intangible situational pressure to the respondents, which may lead the respondents to tick options that do not reflect the actual situation. A total of 600 questionnaires were distributed, and 581 valid questionnaires were returned. Among the valid questionnaires, 51.3% were returned by men and 48.7% by women. Moreover, the age distribution of the respondents ranged from 18–25, 26–35 and 36–45 years old, accounting for 15.8%, 38.7% and 36.1% of the sample, respectively. The respondents' education level mainly reached the undergraduate level, with 49.7% of the sample in that category. The marital status of the respondents was either married or unmarried, with 62.3% who were married and 37.7% who were not. Most respondents were career workers, employees of private enterprises and business operators, representing 19.3%, 38.4%

and 16.8% of the sample, respectively. Finally, the cumulative time respondents spent in the hotels that were included in the study ranged from 3–5 days to 6–8 days, with each timeframe accounting for 42.7% and 32.5% of the sample, respectively.

## 4 Analysis of results

### 4.1 Quantitative results

This study used SPSS 24.0 and AMOS 24.0 to conduct validated factor analysis to test the reliability and validity of the variables to ensure the goodness of fit of the structural model. The Process 3.3 plug-in was also used to test the moderating effect of people's pro-environmental behavior at home.

As the variables involved in this study were all based on customer ratings, it was necessary to test the CMV(common method variance) with Harman's one-way test. The results showed that all factors with eigenvalues greater than 1.0 were more than one, while the maximum factor variance explained was 40.666% and 20.923%, respectively, which were both lower than the benchmark value of 50.00% [59]. Therefore, there was no serious common method bias in this study.

**4.1.1 Measurement model testing.** The reliability of the questionnaire was tested by Cronbach's α and CR. According to the results of the measurement model in Table 1, the Cronbach's α coefficient was greater than 0.7, and the CR was greater than 0.7, representing adequate reliability and internal consistency of the dimensions [60]. The validity measures were examined by convergent validity and discriminant validity, where convergent validity was mainly reflected by the standardized factor loadings, t values and AVE. The results showed that both sets of standardized factor loadings were greater than 0.6 and significant, and the AVE was greater than 0.5, indicating high convergent validity [61].

At the same time, the correlation coefficient between any two variables was less than the square root of the AVE of each variable (Table 2), so the scale had good discriminant validity [61].

**4.1.2 Structural model testing.** After the necessary model corrections, the best-fit model parameters were $\chi 2/df = 2.697 < 3$, RMSEA = $0.054 < 0.08$, GFI = $0.964 > 0.9$, and TIF = $0.960 > 0.9$, which indicated that the fit of both models was good and met the model fit requirements [62].

The results of the hypothesis test (Table 3) indicated that all three dimensions of the hotel home experience had a significant positive effect on pro-environmental behaviors in hotels, with the standardized path coefficients being H1 ($\beta = 0.107$; $p < 0.01$), H2 ($\beta = 0.429$; $p < 0.001$), and H3 ($\beta = 0.220$; $p < 0.001$). The hypotheses were all supported. The functional dimension of home had the greatest influence; the imaginary dimension of home had the second greatest influence, and the emotional dimension of home had the smallest influence.

In addition, the moderating effect of pro-environmental behaviors at home was assessed using the Process 3.3 plug-in. The results showed that people's pro-environmental behaviors at home had a positive moderating effect on the relationship between the functional, emotional and imaginative dimensions of home and people's pro-environmental behaviors in the hotel (Figs 2–4), where $\beta 4a = 0.056$ ($p < 0.05$), $\beta 4b = 0.070$ ($p < 0.01$) and $\beta 4c = 0.054$ ($p < 0.05$). 0.054 ($p < 0.05$). Therefore, H4a, H4b and H4c were all accepted.

### 4.2 Qualitative interpretation

The results of the linear regression analysis showed that the different dimensions of the home experience exerted different degrees of influence on the pro-environmental behaviors of customers in the hotel context. In order to further corroborate the results of the quantitative

**Table 1. Results of the measurement model evaluation.**

| Variables and measurement items | t value | Standard factor loading | Cronbach's α | CR | AVE |
|---|---|---|---|---|---|
| Standards | \|t\|>1.96 | >0.6 | >0.7 | >0.7 | >0.5 |
| **The function of home** | | | 0.972 | 0.972 | 0.851 |
| The hotel rooms are equipped with advanced facilities and are fully functional and easy to use. | | 0.918 | | | |
| The hotel offers different functional spaces, such as a swimming pool, gym, meeting rooms, etc. | 40.338 | 0.931 | | | |
| The physical environment of the hotel matches my tastes. | 37.630 | 0.910 | | | |
| The hotel provides items that match my personal preferences (e.g., pillows, fruits, etc.). | 39.019 | 0.921 | | | |
| The hotel is well-served by the surrounding area and is easily accessible. | 40.603 | 0.932 | | | |
| The daily management of the hotel is strict and protects the privacy and security of the customers. | 39.152 | 0.922 | | | |
| **The emotion of home** | | | 0.883 | 0.872 | 0.534 |
| The hotel provides prompt, proactive and welcoming services. | | 0.706 | | | |
| The hotel takes an active interest in my stay. | 16.926 | 0.766 | | | |
| When I ask for help, the hotel delivers effective solutions. | 16.956 | 0.768 | | | |
| The hotel provides personalized services according to my needs. | 15.195 | 0.683 | | | |
| The hotel organizes a variety of recreational or meaningful activities. | 17.056 | 0.773 | | | |
| Attending events organized by the hotel enhances my relationship with others. | 17.179 | 0.680 | | | |
| **The imagination of home** | | | 0.944 | 0.944 | 0.739 |
| The hotel offers green spaces for relaxation that are soothing. | | 0.858 | | | |
| The hotel has a beautiful setting and a view, and I feel the beauty of the earth's home. | 29.174 | 0.885 | | | |
| The hotel has its own unique design style and cultural ambience and feels like a home with character. | 31.554 | 0.921 | | | |
| The hotel is a home away from home with local flora, scenery or architecture that can be seen. | 29.214 | 0.886 | | | |
| The hotel offers locally relevant items and foods, which make me feel I am staying in a local's home. | 24.771 | 0.809 | | | |
| During important festivals, the hotel creates a strong festive atmosphere and gives me the feeling of being at home. | 23.852 | 0.791 | | | |
| **Pro-environmental behaviors at home** | | | 0.980 | 0.976 | 0.872 |
| When bathing at home, I am careful to control the amount of water I use. | | 0.862 | | | |
| At home, I turn off the lights when I leave the room. | 30.473 | 0.886 | | | |
| At home, I sort my rubbish according to whether it can be recycled. | 33.282 | 0.923 | | | |
| At home, I look for ways to reuse old things. | 34.782 | 0.941 | | | |
| At home, I encourage my family to reduce waste and save water and energy. | 39.857 | 0.992 | | | |
| At home, I prioritize buying green and organic food. | 39.708 | 0.991 | | | |
| **Pro-environmental behaviors in the hotel** | | | 0.920 | 0.924 | 0.636 |
| Whenever I leave the hotel room, I turn off the lights and air conditioning. | | 0.882 | | | |
| When taking a shower in the hotel, I try to control the amount of water I use. | 27.871 | 0.850 | | | |
| I reduce the use of paper (e.g., writing paper, toilet paper, etc.) in this hotel. | 28.500 | 0.860 | | | |
| At this hotel, I reuse bed linen and towels repeatedly. | 27.494 | 0.844 | | | |
| At this hotel, I separate waste according to whether it can be recycled. | 24.643 | 0.795 | | | |
| I encourage others to reduce waste and save water and energy in the hotel. | 18.093 | 0.653 | | | |
| I give preference to organic food in the hotel. | 18.610 | 0.666 | | | |

Note. CR = composite reliability, AVE = average variance extracted.

analysis and to explore the mechanisms involved, 12 of the respondents were selected for semi-structured interviews, and the results of the analysis are as follows.

In the hotel context, the functional dimension of home had only a weak effect on pro-environmental behaviors ($\beta = 0.107$), which was the result of the combined effect of home and nonhome experiences. On the one hand, with regard to the home experience, the ideal home constructed by luxury hotels reinforced customers' functional dependence, while the green

**Table 2. Results of the discriminant validity test.**

| Variables | The function of home | The emotion of home | The imagination of home | Pro-environmental behavior in the hotel |
|---|---|---|---|---|
| The function of home | 0.922 | | | |
| The emotion of home | 0.013 | 0.730 | | |
| The imagination of home | 0.079 | 0.266 | 0.860 | |
| Pro-environmental behavior in the hotel | 0.130 | 0.489 | 0.343 | 0.798 |

practices that satisfied customers' environmentalist identity contributed to these customers engaging in pro-environmental behaviors. On the other hand, with regard to nonhome experiences, the commercial nature of the hotels reinforced customers' perception of otherness within hotel spaces; therefore, at home, individuals may adopt pro-environmental behaviors based on utility and self-interest [63], but when staying at a hotel, individuals are not driven by the same factors. Indeed, as one interviewer reflected, "*The cost of a night in a hotel is fixed, and even if I leave the air conditioning on when I go out, it does not increase the cost of my stay, but I feel comfortable once I get back to my room, so why not*?" Meanwhile, in the tourism context, in which hedonism is prevalent [64], customers are concerned with getting a more heterogeneous and enjoyable experience, ignoring or even opposing green practices, as these practices may lessen the quality of the tourism experience; moreover, the commitment of luxury hotels to create a luxurious and comfortable experience for guests reinforces the idea of the hotel as a hedonistic space and further discourages customers from acting pro-environmentally. Indeed, as one interviewee mentioned, "*Staying in a five-star hotel is about enjoyment, and if I am always thinking about how to save resources and recycle my waste, the experience will be much less enjoyable, and I might as well stay in another hotel that offers better value for the money*."

Second, the emotional dimension of home had a significant effect on pro-environmental behaviors in the hotel context ($\beta = 0.429$), and the results of the interviews showed that although customers may have experienced the emotion of not being at home, this experience did not prevent pro-environmental behaviors. The reason for this result was that prior to their stay, customers saw the hotel as a temporary home and had low expectations with regard to the emotional experience the hotel would provide. As one respondent asserted, "*Unexpectedly,*

**Table 3. Hypothesis testing results.**

| Hypothetical relationships | Path Coefficients | T Values | P Values | Decisions |
|---|---|---|---|---|
| Main Model | | | | |
| H1: FH→PEB in the hotel | 0.107** | 2.828 | 0.005 | Supported |
| H2: EH→PEB in the hotel | 0.429*** | 9.575 | 0.000 | Supported |
| H3: IH→PEB in the hotel | 0.220*** | 5.413 | 0.000 | Supported |
| Moderating Effect of PEBP | | | | |
| PEB in the privacy of the home→PEB in the hotel | 0.334*** | 8.515 | 0.000 | — |
| H4a: FH*PEB at home→PEB in the hotel | 0.056* | 2.493 | 0.013 | Supported |
| H4b: EH*PEB at home→PEB in the hotel | 0.070** | 2.569 | 0.010 | Supported |
| H4c: IH*PEB at home→PEB in the hotel | 0.054* | 2.115 | 0.035 | Supported |

Notes: ① *** $p<0.001$

** $p<0.01$

* $p<0.05$

② FH = The function of home; EH = The emotion of home; IH = The imagination of home; Pro-environmental behavior = PEB

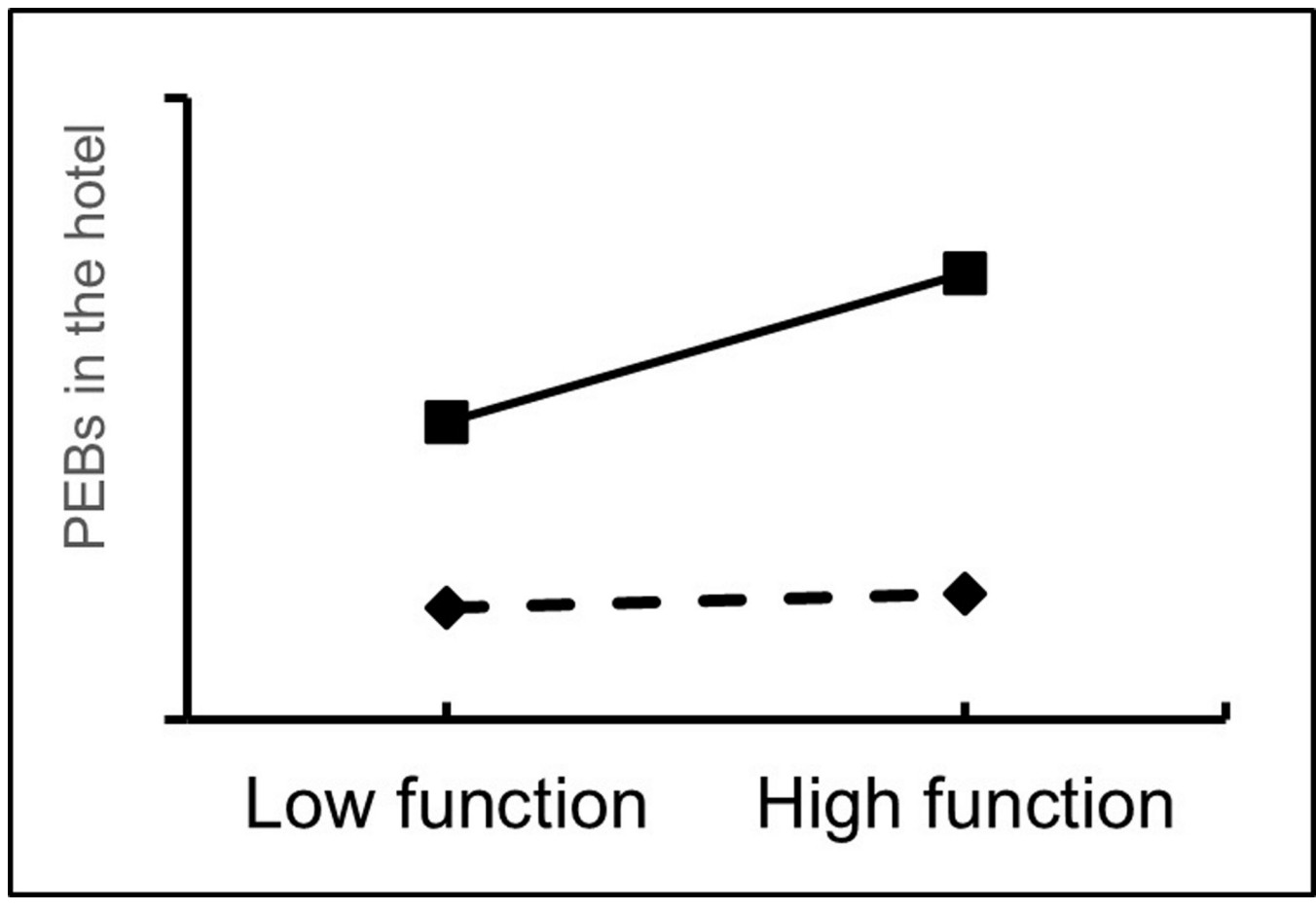

**Fig 2. Moderating effects of PEBs at home (functional dimension of home).**

*the butler gave us home-like care and attention, and the experience was great. This hotel is another home for my holiday, and I will choose to stay there next time.*" At the same time, both satisfaction and positive emotions can promote pro-environmental behavior among customers [10, 39], and this interviewer also mentioned that "I like this hotel so much that I would even protect it like my own home, and I consciously pick up plastic trash when I see it in the coconut grove and on the beach."

Third, the dimension of the imagination of home had a significant effect on customer pro-environmental behaviors in the hotel context (β = 0.220), and the interview results showed that this dimension had a greater effect on customer pro-environmental behaviors, not only enhancing pro-environmental behaviors in the hotel space but also promoting pro-environmental behavioral intentions in everyday life. The green and cultural atmosphere created by the hotel connects me to different scales of places, such as my home, my country and the world. An interviewee said that "*The White Swan Hotel is the first five-star hotel in China, a product of the era of reform and opening up, and a symbol of achieving the rejuvenation of the Chinese nation, where patriotic feelings arise. . . . ..*". Hence, the hotel not only represents the person's home but also embodies the person's hometown, distant home or ideal home. An interviewer also said that "*The hometown water in the lobby of the White Swan Hotel reminds me of my childhood home, where the water is sweet and the moon is bright . . .*". It was clear that the imagination of home featured in the hotel brought up a collection of emotions connected

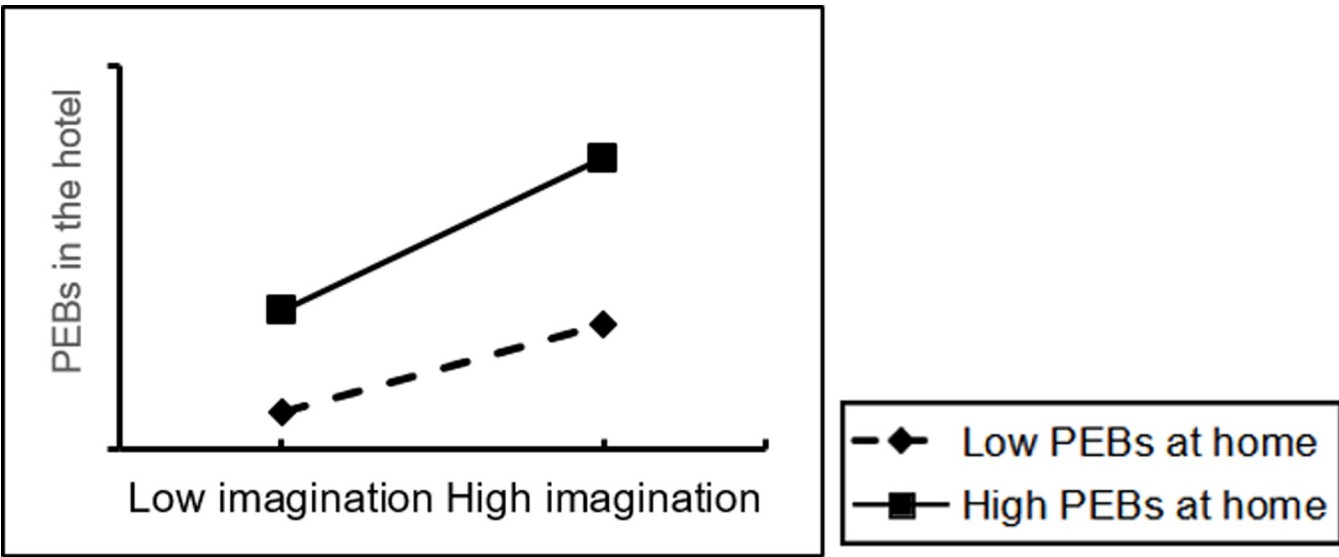

**Fig 3. The moderating effect of PEBs at home (the emotional dimension of home).**

**Fig 4. The moderating effect of PEBs at home (the imaginary dimension of home).**

to the idea of home at different scales, and emotions were the key factors stimulating pro-environmental behavioral intentions. As one interviewer said, "*The lotus flowers in the hotel lake remind me of the lotus flowers on the pond in my 'hometown,' which are beautiful in summer, but due to environmental pollution caused by indiscriminate mining, there is only a small corner of lotus flowers now, when I back home, I want to give some advice to the village about protecting the ecological environment of the countryside.*"

# 5 Conclusions and implications

## 5.1 Conclusions

This paper uses four luxury hotels in China as case studies and adopts a mixed method approach to explore the relationship between people's home experience in hotels and pro-environmental behaviors. The main findings are as follows.

First, the qualitative research shows that people's experience of home in hotels includes three main dimensions, namely, the function of home, the emotion of home and the imagination of home. On the one hand, these three dimensions echo the literature on the experience of home through mobility, i.e., the ways in which people experience home while engaged in caravan tourism, which Wen and Zhang [24] studied. On the other hand, in this study, the different dimensions of the home experience differed according to whether people were engaged in hotel or caravan tourism. With regard to the function of home, caravan travelers focused on how complete the facilities and equipment were in the caravan, while hotel customers focused on a wider geographical scale, evaluating not only the facilities, equipment and management within the hotel but also the hotel's surrounding area. Regarding their emotional interaction of home, caravan travelers mainly targeted their home companions and outsiders, while hotel customers mainly interacted emotionally through their home companions and hotel staff, and had less interaction with other hotel customers. In addition, the stimulus for imagining home differed, with caravan tourists focusing mainly on the natural environment 'outside' and the freedom and familiarity 'inside.' Hotel guests focused on the spaces 'inside,' as the hotel itself had a larger geographical space that not only integrated with the natural environment but also enabled the production of various spaces, such as green spaces, and created a cultural ambience.

Second, the quantitative study showed that the three dimensions of people's experience of home in hotels (based on the function of home, the emotion of home and the imagination of home) all had a significantly positive effect on customers' pro-environmental behaviors. However, the effects of the different dimensions differed, with coefficients of 0.107, 0.429 and 0.220, respectively. The results of this study partially echo existing research claiming that the experience of physical environment [2], subject-object interaction [65], and emotional meaning [66] promotes the production of individual pro-environmental behaviors, but they differ in that this study also confirms the positive effect of spatial imagery on individual pro-environmental behaviors. To further support the relationship between the two dimensions and to explore the mechanisms involved, the study applied semi-structured interviews, which resulted in the finding that the functional and emotional dimensions of home had a joint effect on both home and nonhome. The difference was that for the functional dimension, the home experience had a positive effect, while the nonhome experience had a negative effect, resulting in a weaker effect on the customers' pro-environmental behaviors. However, the emotional dimension of home, the contrast between the expectation of nonhome spaces and the positive emotion regarding home when they stayed in the hotel not only did not weaken their home experience but also made them more satisfied with the hotel, which in turn had a significant positive impact on their pro-environmental behavior. It is worth noting that the results of the

interviews revealed that the imagination of home had a broader scale of influence on pro-environmental behaviors, i.e., it also positively influenced customers' pro-environmental behavioral intentions in their daily lives.

Third, customers' pro-environmental behaviors at home had a positive moderating effect on the relationship between all dimensions of their home experience in hotels and their pro-environmental behavior. On the one hand, although studies have confirmed the moderating effect of past behavior on hotel patrons' pro-environmental behaviors [67], the past behavior thus referred to does not cover the behavior within private homes, which is more closely related to that in hotels. On the other hand, the literature has suggested that the interaction between contextual similarity, self-identity, etc., and pro-environmental behaviors in one context can facilitate the occurrence of pro-environmental behaviors in other contexts [68], and this study echoes this view. Indeed, in the hotel context, the function of home, the emotion of home, and the imagination of home all referred to the intimate sphere. Private home was the object of reference, and there were similarities between the two 'homes.' However, the three dimensions were metaphors for the identities of individuals, and contextual similarity and self-identity interact with private family pro-environmental behaviors, which in turn promote pro-environmental behaviors within the hotel.

## 5.2 Theoretical contributions

This study makes three main theoretical contributions.

First, this study adopts qualitative research methods to construct a conceptual framework to assess people's home experience in the hotel context, which expands the connotations of tourism experience and has theoretical implications for related studies by domestic scholars. First, according to existing research, the tourism experience focuses on the individual's experience of the physical environment [39] and the material and immaterial products [69] of the tourism place, including recreational, ecological, educational and aesthetic experiences [10]. However, hotels represent a special tourism context and a "commercial home" space [33]; therefore, if we simply apply the general tourism experience to explore the experience of home in hotels, it is likely to bias the research, and this study aims to contribute to the connotation of the tourism experience. Second, the existing studies on the tourism experience tend to regard tourist places as real physical containers or backgrounds while ignoring the fact that tourist places are spaces with multiple dimensions, especially the "commercial home" space of hotels, which is not only a physical and emotional space, but also serves as an imaginary space. Therefore, the dimensions of the hotel home experience as derived from this study include not only the material and emotional aspects of home, but also the imagination of home. Finally, the results of this study show that all three dimensions of people's home experience in hotels are reliable and valid, indicating that the scale of the study can be expanded and applied to subsequent studies.

Second, this study extends the empirical research on the relationship between tourism destination experiences and pro-environmental behavior. In the existing related studies, the importance of a destination's green physical environment in motivating pro-environmental behavior is mainly emphasized [10, 53], but the lack of attention to non-green hardware amenities in those works is remedied by this paper. It was found that non-green, home-like functional experiences in a hotel context can likewise promote pro-environmental behaviors among customers. At the same time, existing studies have mainly explored the relationship between the tourism experience and pro-environmental behavior from the perspective of individuals' physical and emotional experiences of physical space, but rarely from the perspective of individuals' experiences of non-physical space (imaginary space). This study finds that, in

the hotel context, customers' experience of non-physical space (imaginary space) has a greater impact on their pro-environmental behavior than their experience of physical space, and it has a positive impact on pro-environmental behavior not only inside the hotel, but also outside the hotel (private home space), thus further expanding the research on "tourism experience - pro-environmental behavior." This further expands the research related to "tourism experience - pro-environmental behavior."

Third, this study links people's pro-environmental behaviors in hotels and at home, addressing the limitations of previous studies that used a binary perspective to explore individuals' pro-environmental behaviors in hotels or at home [58]. In fact, there are similarities between home spaces created by hotels and people's home spaces in terms of the form and the emotion these spaces trigger. Therefore, this finding provides a precondition for people's pro-environmental behaviors to spill over into the home space and allows for the inference that people's pro-environmental behaviors in hotels are more closely related to their intimate home than they are to other tourism contexts. Therefore, when exploring people's pro-environmental behaviors in hotels, it is important to focus on the effects of these pro-environmental behaviors on people's behaviors at home.

### 5.3 Practical implications

First, to enhance hotel customers' pro-environmental behaviors, hotels should not overly pursue a luxurious physical space when seeking to create a home experience. On the one hand, hotels should focus on offering opportunities for green activities. In the era of the experience economy, customers' needs are not only purely based on luxury experiences but also include green experiences. At the same time, studies have shown that green practices in hotels can increase customer satisfaction and loyalty [70, 71], which can also be interpreted as enhancing the customers' sense of a home experience. On the other hand, the emotional dimension of home has the most significant impact on customers' pro-environmental behaviors as they experience home in a hotel. Hotels can also create opportunities or activities with people considered to be from outside the home (other hotel customers) who have common interests and preferences.

Second, linking hotels to other spatial and temporal scales of home enhances the spillover effect of people's hotel-based pro-environmental behaviors. According to the findings of the study, when customers' experience hotel spaces as home, they are inspired to imagine their homes across different spatial and temporal scales, which enhances their intention to engage in pro-environmental behaviors at different geographical scales. This finding shows that hotels can fulfil their role as environmental educators by incorporating local traditions and traditional landscapes and promoting knowledge about the Earth's environment in their construction of 'homes' that embody local, distant and global homes.

### 5.4 Research limitations and future research directions

There are a number of limitations to this study. Indeed, some areas require further attention. This study explored people's experience of home at the scale of a hotel. Although this scale passed the reliability test, the study only focused on the luxury hotel sector; therefore, the applicability of the study to other types of accommodations (e.g., budget hotels, B&Bs, etc.) remains to be investigated. In addition, this study explored the moderating effect of pro-environmental behaviors at home. However, the question of whether a temporal correlation exists between people's pro-environmental behaviors in homelike hotels and people's homes arises. Moreover, the mechanisms involved constitute an area where further research could be developed.

## Supporting information

**S1 Data.**
(XLS)

## Author Contributions

**Conceptualization:** Meixin Liu, Xingxing Peng.

**Data curation:** Xingxing Peng.

**Formal analysis:** Xingxing Peng.

**Funding acquisition:** Meixin Liu.

**Methodology:** Meixin Liu, Xingxing Peng.

**Software:** Xingxing Peng.

**Supervision:** Meixin Liu.

**Visualization:** Xingxing Peng.

**Writing – original draft:** Xingxing Peng.

**Writing – review & editing:** Meixin Liu.

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
