## [Decision Letter · Decision Letter 0]

8 Jan 2023

PONE-D-22-28048Can the home experience in luxury hotels promote pro-environmental behaviors among guests?PLOS ONE

Dear Dr. Peng,

Thank you for submitting your manuscript to PLOS ONE. After careful consideration, we feel that it has merit but does not fully meet PLOS ONE’s publication criteria as it currently stands. Therefore, we invite you to submit a revised version of the manuscript that addresses the points raised during the review process.

We look forward to receiving your revised manuscript.

Kind regards,

You-Yu Dai

Academic Editor

PLOS ONE

Journal Requirements:

2. You indicated that ethical approval was not necessary for your study. Could you please provide further details on why your study is exempt from the need for approval and confirmation from your institutional review board or research ethics committee (e.g., in the form of a letter or email correspondence) that ethics review was not necessary for this study? Please include a copy of the correspondence as an ""Other"" file.

“Financial support recipient: Peng Xingxing

Source of funding: The present study was funded by National Natural Science Foundation of China (Grant No. 42101232);  Natural Science Foundation of Guangdong Province (Grant No. 2022A1515010740).”

Additional Editor Comments:

According to the reviewers' reports, I invite you to do a major revision.

Reviewers' comments:

Reviewer's Responses to Questions

**Comments to the Author**

1. Is the manuscript technically sound, and do the data support the conclusions?

Reviewer #1: Yes

Reviewer #2: Partly

2. Has the statistical analysis been performed appropriately and rigorously? 

Reviewer #1: Yes

Reviewer #2: Yes

3. Have the authors made all data underlying the findings in their manuscript fully available?

Reviewer #1: Yes

Reviewer #2: Yes

4. Is the manuscript presented in an intelligible fashion and written in standard English?

Reviewer #1: Yes

Reviewer #2: Yes

5. Review Comments to the Author

Reviewer #1: This is an interesting study and from an interesting context

I have the following few comments.

- The contribution of the study should be better highlighted in the introduction section

- I believe that the LR could have been more critical in nature

-Figure 1: the conceptual model requires more discussions and justifications

- The Moderating Effect of PEBP requires additional theoretical discussions

- Maybe the author could discuss more on the limitation of the methods used (both the Qualitative and the Quan)

- the practical and theoretical implications of the study would benefit from more discussion

- to cross check typos and references

Reviewer #2: The topic is very interesting and relevant.

Some recommendations to improve the manuscript:

1. in the abstract, present the results found in a more objective and clear way;

2. in the introduction and literature review sections, explain the concept of luxury hotel and explain the option for this type of hotel, as one of the key words of the paper is "luxury hotel" and this reference is made in the title of the article. The words "luxury hotel" first appear in the manuscript at line 245 in research methodology.

3. on line 230 and 231, author said "In this study, mixed methods were used, integrating qualitative and quantitative data to gain a deeper understanding of the research questions". However, this manuscript does not have research questions.

4. in research methodology section a more detailed explanation about the option to carry out a single focus group included six hotel guests + three professors in tourism management + two hotel managers would be necessary, as well as the duration ( minutes) of this focus group and the way in which data were recorded (audio?).

5. at the endof analysis of results section, a comparison with results of studies from other academic literature will be necessary.

6. PLOS authors have the option to publish the peer review history of their article (what does this mean?). If published, this will include your full peer review and any attached files.

Reviewer #1: No

Reviewer #2: No

---

## [Author Response · Author response to Decision Letter 0]

22 Feb 2023

Reviewer 1

1.The contribution of the study should be better highlighted in the introduction section.

Respond: Thank you very much for your feedback. In the introduction section, we have described the contribution of this study in more specific terms.

Please see details on page 3-4 with highlighted text. 

2.I believe that the LR could have been more critical in nature.

Respond: Thank you for the constructive suggestion. We have added a more critical discussion of luxury hotels. 

Please see the detailed explanation highlighted on page 2.

3.Figure 1: the conceptual model requires more discussions and justifications.

Respond: Thank you for pointing this out. We have added a more specific explanation to this conceptual model.

Please see details on page 7 with highlighted text.

4.The Moderating Effect of PEBP requires additional theoretical discussions.

Respond: Thank you for your suggestion. We have added a theoretical exploration of the moderating effect of PEBP, thus making the hypothesis more convincing.

Please see details on page 6-7 with highlighted text.

5.Maybe the author could discuss more on the limitation of the methods used (both the Qualitative and the Quan).

Respond: Thank you for pointing this out. In the research methods section, we have added further discussion on the limitations of qualitative and quantitative research and the reasons for using a mixed research approach in this study.

Please see the first paragraph on page 8 for specific details with highlighted text.

6.the practical and theoretical implications of the study would benefit from more discussion.

Respond: Thank you for the constructive suggestion. We have revised the theoretical and practical implications section, especially emphasizing the theoretical implications, which we hope will lead to a more in-depth discussion of this study.

Please see details on page 16-17 with highlighted text.

7.to cross check typos and references.

Respond: Thank you for your suggestion. We apologize for any inconvenience to your review, and we have double-checked for typos and references.

Reviewer 2

1. in the abstract, present the results found in a more objective and clear way;

Respond: Thank you very much for your feedback. We have removed any redundant sentences to make the summary more concise.

Please see details on page 2 with highlighted text. 

2. in the introduction and literature review sections, explain the concept of luxury hotel and explain the option for this type of hotel, as one of the key words of the paper is "luxury hotel" and this reference is made in the title of the article. The words "luxury hotel" first appear in the manuscript at line 245 in research methodology.

Respond: Thank you for your suggestion. In the introduction section, we have added the concept of luxury hotels and a critical discussion of their roles.

Please see details on page 2 with highlighted text. 

3. on line 230 and 231, author said "In this study, mixed methods were used, integrating qualitative and quantitative data to gain a deeper understanding of the research questions". However, this manuscript does not have research questions.

Respond: Thank you for pointing this out. Regarding "In this study, mixed methods were used, integrating qualitative and quantitative data to gain a deeper understanding of the research questions"，our intention was to explain the reasons for adopting a mixed research approach in this paper, and to clarify the limitations of using qualitative and quantitative research.

In addition, the core research question of this paper is "whether the home experience has an impact on the pro-environmental behavior of luxury hotel customers", and the choice of research method is specifically meant to address this question.

Please see the first paragraph on page 8 for specific details with highlighted text.

4. in research methodology section a more detailed explanation about the option to carry out a single focus group included six hotel guests + three professors in tourism management + two hotel managers would be necessary, as well as the duration (minutes) of this focus group and the way in which data were recorded (audio?).

Thank you for your suggestion. We have added the member presentations and data acquisition process for the focus group discussions.

Please see the second paragraph on page 8 for specific details with highlighted text.

5. at the end of analysis of results section, a comparison with results of studies from other academic literature will be necessary.

Respond: Thank you for your constructive suggestion. We have revised the conclusion by adding a dialog discussion of the existing relevant literature.

Please see details on page 15-16 with highlighted text.

---

## [Decision Letter · Decision Letter 1]

13 Apr 2023

Can the home experience in luxury hotels promote pro-environmental behaviors among guests?

PONE-D-22-28048R1

Dear Dr. Xingxing Peng,

We’re pleased to inform you that your manuscript has been judged scientifically suitable for publication and will be formally accepted for publication once it meets all outstanding technical requirements.

Kind regards,

You-Yu Dai

Academic Editor

PLOS ONE

Additional Editor Comments (optional):

Reviewers' comments:

Reviewer's Responses to Questions

**Comments to the Author**

1. If the authors have adequately addressed your comments raised in a previous round of review and you feel that this manuscript is now acceptable for publication, you may indicate that here to bypass the “Comments to the Author” section, enter your conflict of interest statement in the “Confidential to Editor” section, and submit your "Accept" recommendation.

Reviewer #1: All comments have been addressed

Reviewer #2: All comments have been addressed

2. Is the manuscript technically sound, and do the data support the conclusions?

Reviewer #1: Yes

Reviewer #2: Partly

3. Has the statistical analysis been performed appropriately and rigorously? 

Reviewer #1: Yes

Reviewer #2: Yes

4. Have the authors made all data underlying the findings in their manuscript fully available?

Reviewer #1: Yes

Reviewer #2: Yes

5. Is the manuscript presented in an intelligible fashion and written in standard English?

Reviewer #1: Yes

Reviewer #2: Yes

6. Review Comments to the Author

Reviewer #1: i am overall satisfied with this improved version of the paper. The author has addressed most of the comments

Reviewer #2: Many congratulations to the authors for the excellent job of revising and improving their manuscript.

Weel done!

7. PLOS authors have the option to publish the peer review history of their article (what does this mean?). If published, this will include your full peer review and any attached files.

Reviewer #1: No

Reviewer #2: No

---

## [Editor Report · Acceptance letter]

15 Aug 2023

PONE-D-22-28048R1 

Can the home experience in luxury hotels promote pro-environmental behavior among guests? 

Dear Dr. Peng:

I'm pleased to inform you that your manuscript has been deemed suitable for publication in PLOS ONE. Congratulations! Your manuscript is now with our production department. 

Kind regards, 

on behalf of

Dr. You-Yu Dai 

Academic Editor

PLOS ONE